# The Importance of CXCL1 in the Physiological State and in Noncancer Diseases of the Oral Cavity and Abdominal Organs

**DOI:** 10.3390/ijms23137151

**Published:** 2022-06-28

**Authors:** Jan Korbecki, Iwona Szatkowska, Patrycja Kupnicka, Wojciech Żwierełło, Katarzyna Barczak, Iwona Poziomkowska-Gęsicka, Jerzy Wójcik, Dariusz Chlubek, Irena Baranowska-Bosiacka

**Affiliations:** 1Department of Biochemistry and Medical Chemistry, Pomeranian Medical University in Szczecin, Powstańców Wlkp. 72 Av., 70-111 Szczecin, Poland; jan.korbecki@onet.eu (J.K.); patrycja.kupnicka@pum.edu.pl (P.K.); dchlubek@pum.edu.pl (D.C.); 2Department of Ruminants Science, Faculty of Biotechnology and Animal Husbandry, West Pomeranian University of Technology, Klemensa Janickiego 29 St., 71-270 Szczecin, Poland; iwona.szatkowska@zut.edu.pl (I.S.); jerzy.wojcik@zut.edu.pl (J.W.); 3Department of Medical Chemistry, Pomeranian Medical University in Szczecin, Powstańców Wlkp. 72 Av., 70-111 Szczecin, Poland; wojciech.zwierello@pum.edu.pl; 4Department of Conservative Dentistry and Endodontics, Pomeranian Medical University, Powstańców Wlkp 72, 70-111 Szczecin, Poland; katarzyna.barczak@pum.edu.pl; 5Clinical Allergology Department, Pomeranian Medical University in Szczecin, Powstańców Wlkp. 72 Av., 70-111 Szczecin, Poland; iwona.poziomkowska@op.pl

**Keywords:** CXCL1, CXCR2, chemokine, cytokine, neutrophil, MIP-2, CINC-1, KC, Gro-α

## Abstract

CXCL1 is a CXC chemokine, CXCR2 ligand and chemotactic factor for neutrophils. In this paper, we present a review of the role of the chemokine CXCL1 in physiology and in selected major non-cancer diseases of the oral cavity and abdominal organs (gingiva, salivary glands, stomach, liver, pancreas, intestines, and kidneys). We focus on the importance of CXCL1 on implantation and placentation as well as on human pluripotent stem cells. We also show the significance of CXCL1 in selected diseases of the abdominal organs, including the gastrointestinal tract and oral cavity (periodontal diseases, periodontitis, Sjögren syndrome, *Helicobacter pylori* infection, diabetes, liver cirrhosis, alcoholic liver disease (ALD), non-alcoholic fatty liver disease (NAFLD), HBV and HCV infection, liver ischemia and reperfusion injury, inflammatory bowel disease (Crohn’s disease and ulcerative colitis), obesity and overweight, kidney transplantation and ischemic-reperfusion injury, endometriosis and adenomyosis).

## 1. Introduction

CXC motif chemokine ligand 1 (CXCL1) is a chemokine that belongs to the CXC chemokines, one of four sub-families of fifty chemotactic cytokines divided according to the N-terminal cysteine motif; the characteristic feature of this subfamily is a conserved CXC motif that forms two disulfide bridges [1,2]. Although CXCL1, also known as melanoma growth-stimulatory activity (MGSA) and growth-regulated (or -related) oncogene-α (Gro-α), is one of the most studied CXC motif chemokine receptor 2 (CXCR2) receptor ligands, there is a lack of reviews summarizing the accumulated knowledge about this chemokine. Following our previous paper on the role of CXCL1 in physiology and in bone, muscle and brain diseases [3], in this paper, we present the importance of CXCL1 in other organs and tissues in the physiological state and in noncancer diseases of the oral cavity and abdominal organs (Figure 1).

## 2. Comments on Research Methodology

CXCL1 is a chemokine that belongs to the CXC sub-family of chemokines [1]. The paralog for this chemokine in rats is cytokine-induced neutrophil chemoattractant-1 (CINC-1) [4,5] and murine keratinocyte-derived chemokine (KC) [6,7]. All these chemokines are CXCR2 ligands [8,9,10,11] and chemotactic factors for neutrophils [12,13,14]. Therefore, CINC-1, KC and CXCL1 play an important role in the development of inflammation and their names are used interchangeably when comparing research data between laboratory animals and data obtained from humans. Nevertheless, the properties of CINC-1 and KC are not identical to CXCL1 and cannot be easily compared. The results on a mouse and rat model do not completely reflect results from humans [15,16]. In rodents, there are five CXCR2 ligands compared to seven in humans [1]. In addition, human CXCR2 ligands differ from rat chemokines in the regulation of their expression, which has evolved over the course of evolution. For this reason, some results obtained in animal models over a given chemokine cannot be compared directly to data obtained in humans.

Because of the difficulties described when analyzing the importance of CXCL1 in a given disease, we first showed altered CXCL1 levels in patients. We then assumed that if levels of a chemokine equivalent to human CXCL1 are elevated in laboratory animals with the same disease, then this chemokine has the same role in disease mechanisms as human CXCL1. Nevertheless, in some diseases, animals show elevated levels in the expression of their counterparts of CXCL1, while in humans, we observed no changes. In this case, we indicated that the data cited are from animal studies, and the reported mechanisms may differ in humans with a given disease.

## 3. Gastrointestinal System and Organs Associated with Digestion

### 3.1. Periodontal Diseases and Periodontitis

Periodontal diseases are inflammation diseases of the gingiva [17]. In its advanced stage, periodontal diseases progress to periodontitis, which is associated with the loss of gingiva, alveolar bone and ligaments, and the formation of periodontal pockets, exposing the tooth root. Untreated periodontitis leads to tooth loss. It is estimated that more than 40% of the population of upper-middle-income and high-income countries suffer from periodontitis [18], and only about 9% of adults do not have periodontal disease [18]. Periodontal disease is initiated by microorganisms that form dental plaque [17]. In particular, severe forms of periodontitis are caused by *Aggregatibacter actinomycetemcomitans* and *Porphyromonas gingivalis* [17]. In the initial stages of the disease, inflammation occurs in the gingiva.

In humans with periodontitis, there is an increased amount of CXCL1 in the affected gingiva (Figure 2) [19]. Production of this chemokine in gingival fibroblasts is caused by bacteria such as *Fusobacterium nucleatum* [19] and *Porphyromonas gingivalis* [20,21]. This effect is enhanced by a fatty diet, particularly saturated fatty acids (SFA) [20]. The response of gingival fibroblasts taken from periodontitis patients to lipopolysaccharide (LPS) is much lower than in cells taken from healthy individuals, which is probably related to chronic inflammation [21]. The increased expression of CXCL1 in periodontitis may be caused by other factors such as high mobility group box 1 (HMGB1) [22].

In humans, CXCL1 causes neutrophil recruitment to inflamed sites in the gingiva [23]. In a mouse model of periodontitis, commensal bacteria increase the expression of macrophage inflammatory protein-2 (MIP-2) but not KC, which means that MIP-2 is responsible for neutrophil recruitment in mice [24,25]. Neutrophils cause the recruitment of T helper type 17 (Th17) cells, which in turn induces the recruitment of neutrophils [23]. In the gingival crevice, neutrophils participate in the chronic inflammatory response. They destroy bacteria that cause inflammation but also cause tissue destruction by producing reactive oxygen species (ROS) and secreting various other factors from neutrophil granules. The effect of neutrophils is enhanced in patients with periodontitis because oral neutrophils from patients with this disease show a higher production of ROS [26]. CXCL1 also acts on preosteoclasts [27]. CXCL1 is a chemoattractant for preosteoclasts and causes osteoclastogenesis [28,29]. For this reason, inflammatory reactions during periodontal diseases cause bone resorption and consequent enlargement of periodontal pockets, which leads to tooth loss in the advanced stages of the disease [27].

### 3.2. Sjögren Syndrome

Sjögren syndrome is characterized by inflammatory reactions in exocrine glands, particularly salivary glands and lacrimal glands [30]. The incidence of this autoimmune disease is estimated to be between 3 and 11 cases per 100,000, with the majority being women. The CXCL1→CXCR2 axis plays an important role in this disease, with high expression of CXCL1 in the salivary and lacrimal glands of patients [31]. By activating the CXCR2 receptor, CXCL1 causes an increase in the activity of disintegrin and metalloproteinase 17 (ADAM17) [32], a metalloprotease that causes a release of the amphiregulin, an epidermal growth factor receptor (EGFR) ligand that causes activation of EGFR [33]. This results in an increase in the release of inflammatory cytokines, one of the inflammatory response factors in Sjögren syndrome [31,33].

### 3.3. Helicobacter Pylori Infection, Gastritis and Peptic Ulcer Disease

*Helicobacter pylori* is a Gram-negative bacterium with a spiral morphology and four to six unipolar flagella [34]. The bacterium has an elongated shape and is 2.5 to 5 μm long and 0.5 to 1 μm wide. The genome size of *H. pylori* is approximately 1.67 Mb, with plasmids ranging in size from 1.5 to 23.3 kb. It is estimated that, on average, more than half of the global population carries this microaerophilic bacterium [35], from 24% in Oceania to 87.7% in Nigeria. *H. pylori* is a pathogen that causes dyspepsia, gastritis and peptic ulcer disease [36]. Untreated infection may result in gastric carcinoma and also mucosa-associated lymphoid tissue lymphomas, depending on environmental factors, factors related to the infected individual, and factors affecting the virulence of *H. pylori,* such as cytotoxin-associated gene A (CagA), which in turn is translocated by the type IV secretion system (T4SS) into host cells [36,37]; vacuolating cytotoxin A (VacA), duodenal ulcer promoting gene A (DupA), outer membrane inflammatory protein A (OipA), gamma-glutamyl transpeptidase (GGT) [36,38,39] and *H. pylori*-neutrophil activating protein (HP-NAP) [40].

*H. pylori* infection and its pathogenic activity are associated with inflammatory responses, including an increase in CXCL1 expression, as well as other ligands for CXCR2, in particular CXCL2, CXCL3, CXCL5 and CXCL8/IL-8 in gastric epithelial cells in gastric mucosa [41,42,43,44,45]. *H. pylori* infection also increases CXCL1 and CXCL8/IL-8 expression in macrophages and neutrophils in the gastric epithelium [46,47]. In CD1c^+^ conventional dendritic cells, *H. pylori* induces the expression of CXCL1 and CXCL8/IL-8 [48]. Toll-like receptor 2 (TLR2) activation decreases the expression of these chemokines but increases tumor necrosis factor α (TNF-α) and granulocyte-macrophage colony-stimulating factor (GM-CSF) secretion from CD1c^+^ conventional dendritic cells. *H. pylori* also increases the expression of CXCR2 ligands in gastric cancer cells, which constitutes a potential therapeutic target in gastric cancer [49,50]. The effect of *H. pylori* on CXCR2 ligand expression is mediated by (Figure 3):Virulence factor CagA in gastric epithelial cells [44,49,51]. This factor activates the Ras→mitogen-activated protein kinase kinase (MEK)→extracellular signal-regulated kinase (ERK) mitogen-activated protein kinase (MAPK) pathway [52]. The latter kinase activates nuclear factor κB (NF-κB), which increases the expression of chemokines that are ligands for CXCR2. NF-κB activation by CagA also involves transforming growth factor-beta-activated kinase 1 (TAK1) [53]. At the same time, it seems that NF-κB activation may only indirectly affect the expression of CXCR2 ligands. More specifically, NF-κB activation increases interleukin-32 (IL-32) levels in the cytoplasm, which increases the expression of CXCR2 ligands [51];TLR2 activation on gastric epithelial cells by *H. pylori* [45]. Increased expression of CXCR2 ligands is independent of CapA and toll-like receptor 4 (TLR4) but dependent on TLR2 activation. This is followed by activation of EGFR, ERK MAPK, c-Jun N-terminal kinase (JNK) MAPK and signal transducer of activation (STAT) signaling pathways [45], which increase the expression of CXCR2 ligands. Studies on neutrophils have shown that HP-NAP increases CXCL8/IL-8 secretion via TLR2 activation [54], which indicates that this virulence factor may be involved in increased CXCL1 expression in gastric epithelial cells independent of CagA;NF-κB activation by *H. pylori* secreted TNF-α inducing protein (Tip-α) [55]. This mechanism was demonstrated on mouse gastric cancer MGT-40 cells.

*H. pylori* also increases the expression of CXCR2 in gastric mucosa [56] through the activation and direct action of nuclear factor-κB subunit 1 (NF-κB1). Increased expression of CXCR2 ligands, as well as CXCR2 itself, lead to the activation of this receptor and to senescence of gastric epithelial cells. This process is dependent on p53, and CXCR2 only enhances this process by increasing p53 activity. The senescence of gastric epithelial cells is followed by gastric mucosal atrophy [56]. However, senescence is a process in which p53 inhibits the proliferation of cells [57]; that is, it is a mechanism that protects the body from cancer formation. However, *H. pylori* also increases genomic instability, which leads to mutations in the *TP53* gene [58,59]. With either reduced or absent p53 activity, this process contributes to cancer formation [57], especially as CXCR2 increases cell proliferation in the absence of p53. This mechanism facilitates the formation of gastric cancer.

At the same time, *H. pylori* can also decrease the expression of ligands for CXCR2. In particular, CagA decreases the expression of IL-17 receptor B (IL-17RB) in gastric mucosa [60], a receptor for IL-17 that increases the stability of the CXCL1 transcript [61,62]. Decreasing the expression of IL-17RB decreases the expression of CXCL1 [60].

The increased levels of CXCL1 and other CXCR2 ligands lead to infiltration of gastric mucosa by neutrophils [49]. However, the mentioned chemokines are not the only factors causing neutrophil accumulation in gastric mucosa in *H. pylori* infection. *H. pylori* infection also increases the expression of hepatoma-derived growth factor (HDGF), which causes neutrophil recruitment [63], and HP-NAP, which causes neutrophil adhesion to the endothelium [64] and neutrophil transendothelial migration [65]. The latter virulence factor increases the lifespan of neutrophils by acting on monocytes [40,66]. The aforementioned recruitment processes increase neutrophil accumulation in regions of *H. pylori* infection [67]. Subsequently, neutrophils participate in chronic inflammation associated with *H. pylori* infection. This bacterium causes N1-like subtype differentiation of neutrophils [47]. The bacterium increases interleukin-1β (IL-1β) production in these cells by activating the nucleotide-binding domain and leucine-rich repeat-related (NLR) family, pyrin domain containing 3 (NLRP3) inflammasome [47,68,69]. This process is dependent on T4SS, TLR2 and TLR4 [68,69]. HP-NAP also causes an oxidative burst in neutrophils [70]. The activation of NLRP3 and the oxidative burst lead to chronic inflammation, which has a destructive effect on the gastric mucosa, resulting in dyspepsia, gastritis and peptic ulcer disease.

### 3.4. Diabetes

Diabetes is a group of diseases associated with excessively high blood glucose levels. We distinguish type 1 diabetes mellitus (T1DM) [71] and type 2 diabetes mellitus (T2DM) [72]. T1DM is an autoimmune disease in which autoantibodies destroy pancreatic β cells. As a result, the body’s ability to produce insulin is reduced. The incidence of this type of diabetes is estimated at 22.9 cases per 100,000 [71]. T2DM is caused by insulin resistance of the organs responsible for glucose metabolism, which is associated with hyperinsulinaemia [72]. Abnormal blood glucose levels result in a loss of pancreatic β cells and thus a decrease in insulin production in advanced and untreated T2DM. T2DM is the most common type of diabetes, affecting an estimated 370 million people worldwide. Other types of diabetes include 3c diabetes, previously known as pancreatogenic diabetes mellitus [73]. The main cause of 3c diabetes is chronic pancreatitis, but also pancreatic ductal adenocarcinoma, haemochromatosis and cystic fibrosis, diseases that lead to chronic pancreatic inflammation resulting in β cell dysfunction, β cell loss and fibrosis of the pancreas. It is estimated that 1% to 9% of all diabetes cases are 3c diabetes [73]. There is also gestational diabetes, which is associated with pregnancy and represents pregnancy complications [74]. It is estimated that one in six pregnancies is burdened with gestational diabetes.

CXCL1 is involved in the pathophysiology of T1DM (Figure 4). Patients with T1DM have elevated blood levels of CXCL1 [75]. The level of this chemokine is higher than in T2DM patients and in healthy individuals and is associated with the rate of disease progression [76]. During inflammatory responses in T1DM, pancreatic islets are infiltrated by macrophages, which secrete IL-1β and CXCR2 ligands [77]. Following the IL-1β-induced expression and secretion of CXCR2 ligands by β cells [77,78], CXCR2-dependent neutrophils are recruited to pancreatic islets. Accumulation of neutrophils has been confirmed in the pancreas in patients with T1DM [79,80]. In mouse models, under the influence of activated B-1a cells, neutrophils secrete cathelicidin-related antimicrobial peptide (CRAMP) [81], a mouse peptide that binds to DNA derived from cell debris; the human analog for CRAMP is LL-37. dsDNA-specific IgG secreted by B-1a cells attaches to the CRAMP/LL-37-DNA complex, which activates plasmacytoid dendritic cells (pDCs). This increases the destruction of pancreatic islets in the early stages of T1DM.

The level of CXCL1 is also elevated in T2DM [82] and causes a decrease in islet function [83], which leads to decreased insulin secretion and thus hyperglycemia. Nevertheless, studies in a rat model have shown that CXCR2 ligands do not act directly on β cells [84]. They do not cause apoptosis of these cells or affect insulin secretion, which means that CXCL1 is only indirectly involved in β cell destruction. In T2DM, there is no infiltration of the pancreas by neutrophils, which may indicate that these cells are not involved in pancreatic dysfunction in this disease [79]. However, saturated fatty acids cause inflammation of pancreatic islets [85], which leads to the increased expression of pro-inflammatory cytokines, including CXCL1. This is important because about 60% of the North American and European populations are overweight [86], and obesity is associated with insulin resistance and T2DM [72]. This means that the role of CXCL1 in cell dysfunction in T2DM patients needs to be thoroughly investigated.

Neutrophils are also important in pancreatic islet destruction after transplantation [87,88]. This is a consequence of NF-κB activation in pancreatic islet cells, which results in an increased expression of chemokines responsible for neutrophil recruitment, including CXCL1, CXCL2 and CXCL6.

CXCL1 levels are also elevated in the blood of patients with gestational diabetes mellitus and neonates whose mothers had gestational diabetes mellitus [89]. However, the importance of CXCL1 in gestational diabetes mellitus remains to be investigated.

Diabetes is a multisystem disease, and advanced, untreated diabetes may lead to diabetic ketoacidosis and thus neuroinflammation. Although a study on rats has not shown elevated brain concentrations of the rat equivalent of CXCL1 during diabetic ketoacidosis [90], the link between diabetes, neuroinflammation and CXCL1 needs to be investigated in a human model.

### 3.5. The Physiology of Liver

CXCR2 ligands in the liver play an important physiological role in protecting the body from microbial flora, as demonstrated in a mouse model. Gut microbes contribute to the increased expression of KC in the liver [91], and all liver cells are responsible for the production of this chemokine. However, KC production occurs mainly in hepatic stellate cells (HSCs) in a process dependent on TLR4. KC in the liver causes infiltration of this organ by neutrophils, which allows the protection of this organ through the low-level translocation of bacteria originating from the gut [91]. Nevertheless, the mechanism described needs to be confirmed in a human model. We also do not know which CXCR2 ligand is responsible for protecting the human body from microbial flora.

### 3.6. Liver Diseases, Cirrhosis

Liver diseases include various liver conditions with different etiologies. In this section, we will discuss the role of CXCL1 in the major liver diseases, namely non-alcoholic fatty liver disease (NAFLD) [92], alcoholic liver disease (ALD) [93], and hepatitis B virus (HBV) [94] and hepatitis C virus (HCV) [95] infection. CXCL1 is one of the components that play an important role in these conditions (Figure 5).

NAFLD is caused by an abnormal diet with too much fat and simple sugars [92], which leads to liver inflammation, liver fibrosis and liver steatosis. It is estimated that 20%–30% of people consuming a Western-style diet exhibit NAFLD—a value that will likely rise as the obesity epidemic is only increasing in developed countries [86]. ALD, on the other hand, arises from excessive and heavy alcohol consumption [93]. In advanced stages, ALD progresses to liver steatosis and then to liver cirrhosis and hepatocellular carcinoma. It is estimated that half a million people die annually from alcoholic liver cirrhosis, mostly in North America and Europe [93]. Liver diseases can also be associated with viral infection: HBV [94] and HCV [95]. HBV is a virus belonging to the *Hepadnaviridae* family with partially double-stranded circular DNA of 3.2 kb in length [96]. An estimated 240 million people worldwide are infected with this virus [94], but this number is expected to decrease with the spread of the HBV vaccine. Overall, 40% of those infected develop liver cirrhosis. Another virus that affects the liver, HCV, belongs to the *Flaviviridae* family, which has a positive-sense, single-stranded RNA of 9.6 kb in length [97]. An estimated 177.5 million people worldwide are infected, or about 2.5% of the total population [98]. All the aforementioned liver diseases are associated with liver inflammation and fibrosis, which destroys the tissue structure in this organ. In the advanced stage of the disease, this condition is called liver cirrhosis [99]. It is estimated that liver cirrhosis leads to more than 1 million deaths annually. In addition to cirrhosis, all the aforementioned liver diseases lead to hepatocellular carcinoma [92,93].

In mice, high-fat diet-induced NAFLD results in an increased expression of KC and MIP-2 in the liver [100,101]. This is because SFAs, including palmitic acid, increase NF-κB, ERK MAPK and JNK MAPK–dependent KC expression in murine hepatocytes [102,103]. Research on human patients has shown that CXCL1 is one of the most important hub genes that are associated with NAFLD but also with acute myocardial infarction associated with NAFLD [104].

Patients with ALD show increased expression of CXCL1 in the liver [105,106]. This chemokine can be produced under the influence of ethanol by HSC, as shown by KC expression analyses [106]. A high-fat diet combined with alcohol increases KC expression in HSCs, endothelial cells and most significantly in hepatocytes, as shown by experiments in mice [102]. As KC is a paralog for human CXCL1, the described mechanism may take place in humans. The important result of increased CXCL1 expression is parenchymal neutrophil infiltration, which damages the liver [105]. A high-fat diet potentiates the effect of ethanol on KC expression in mice [102].

Increased CXCL1 expression in the liver can also be caused by HBV or HCV infection. HBV infection is associated with chronic inflammation of the liver, which leads to liver fibrosis. Transforming growth factor β1 (TGF-β1) and CXCL1 play an important role in this process. During HBV infection, TGF-β1 expression is increased in hepatocytes [107] due to the direct attachment of the hepatitis B virus X (HBx) antigen to sequences within the *TGFB1* gene. TGF-β1 causes activation of HSCs [108], which is followed by an increase in the expression of CD147/basigin on HSC; CD147/basigin leads to increased expression and secretion of CXCL1 by these cells [109].

HCV infection causes chronic liver inflammation, which leads to liver cirrhosis. An important element in the course of this disease is CXCL1—its expression is elevated in infected hepatocytes [110]. At the same time, CXCL1 expression is significantly more increased in infected hepatocytes that interact with HSC. During HCV infection, CXCL1 levels are increased in serum [111] and in the liver of patients [112]. The level of CXCL1 expression in the liver is associated with disease progression, stages of liver fibrosis [112] and liver cirrhosis [113]. In particular, in patients that are carriers of the CXCL1 rs4074 A allele, HCV increases CXCL1 expression much more than in patients who are not [113]. As a consequence of this, patients with the CXCL1 rs4074 A allele have a greater predisposition to liver cirrhosis as a result of chronic HCV infection. However, a study by Johansson et al. showed that the ISHAK fibrosis score positively correlates with increased serum CXC motif chemokine ligand 8 (CXCL8)/interleukin-8 (IL-8) levels but negatively with CXCL1 levels [114].

CXCL1 plays an important role in liver diseases. It causes increased expression of α-smooth muscle actin (α-SMA) and alpha-1 type I collagen in HSC [109,115], which leads to liver fibrosis. CXCL1 also causes infiltration of the liver by neutrophils [105,106] which then activate HSC. This causes an increase in the expression of GM-CSF and interleukin-15 (IL-15) in HSC, factors that increase the survival of neutrophils [116]. Neutrophils produce ROS, which increases collagen production by HSCs [117], followed by liver injury and liver fibrosis [116]. Importantly, CXCL1 is not directly cytotoxic to hepatocytes [102]. In contrast, it can increase hepatocyte proliferation, as shown by experiments on mouse hepatocytes [100].

### 3.7. Liver Ischemia and Reperfusion Injury

Liver ischemia and reperfusion injury often result from liver transplantation [118]. This process is caused by the resumption of blood flow to the liver after transplantation, which results in an inflammatory response and damage to the organ. Ischemia and reperfusion in mice are associated with an increase in the expression of TNF-α, KC and MIP-2 in the liver [119,120]; the chemokines KC and MIP-2 are produced by Kupffer cells [119].

Experiments in rats have shown that hepatocytes are the main source of the ligands for CXCR2 [121]. Increased expression of CXCR2 ligands is followed by the infiltration of the liver by neutrophils [119,122,123] which cause liver injury [122,124]. Studies on liver transplantation patients have shown that CXCL1 levels in the blood were not altered [125]. In contrast, liver ischemia and reperfusion were followed by an increase in blood CXCL8/IL-8 levels [125,126]; this chemokine in humans plays the role played by KC and MIP-2 in mice during liver ischemia and reperfusion injury.

Liver transplantation may also be associated with lung injury. Liver transplantation in mice is associated with an increase in lung expression of KC, MIP-2 [127] and CXCL5/ENA-78 [128]. This effect is dependent on TNF-α, which is produced in the liver [128]. Increased expression of CXCR2 ligands in the lung leads to infiltration of the lung by neutrophils, which causes pulmonary injury. More studies of liver transplantation patients are required to confirm and analyze the significance of CXCL1 in lung injury.

### 3.8. CXCL1 as an Acute-Phase Protein (APP)

CXCL1 may be one of the acute-phase proteins (APP). If extensive tissue necrosis occurs in mice, damage-associated molecular pattern molecules (DAMPs) are released into the bloodstream, which leads to an increase in KC expression in many organs such as the spleen, kidney, and heart [129]; however, the greatest increase in the expression of this chemokine can be found in the liver. This process involves Kupffer cells, which secrete TNF-α that increases KC expression in hepatocytes. A similar mechanism occurs following spinal cord injury [130,131] and focal brain injury [132], with an increase in hepatic expression of CINC-1 in rats and KC in mice. Increased expression of these chemokines leads to increased levels of KC or CINC-1 in the bloodstream, which causes neutrophil egress from the bone marrow via CXCR2 [133]. Then, neutrophils are recruited either to sites of inflammatory responses or to sites of tissue damage, where they perform their function. However, an excessive liver response can lead to multiple organ dysfunction syndrome (MODS) [132]. For this reason, there is a mechanism that inhibits the excessive response. In mice, KC, along with serum amyloid A, causes mobilization of myeloid-derived suppressor cells (MDSC) [134], cells that inhibit inflammatory reactions and thus protect against an overly intense body response to tissue damage, extensive inflammatory reactions or LPS in sepsis. Nevertheless, studies on sepsis patients have shown that in humans, the main chemokine involved in the aforementioned mechanism is not CXCL1, but CXCL8/IL-8 [135], whose blood levels are significantly increased in sepsis. CXCL1 levels tend to increase but are only close to statistical significance [135].

### 3.9. Inflammatory Bowel Disease: Crohn’s Disease and Ulcerative Colitis

Inflammatory bowel disease is any disease resulting from inflammatory reactions in the gastrointestinal tract; the most significant are Crohn’s disease [136] and ulcerative colitis [137]. The highest prevalences of inflammatory bowel disease are in North America and in western and northern Europe [138]. The estimated prevalence of Crohn’s disease in North America is 96.3–318.5 per 100,000 people and for ulcerative colitis, 139.8–286.3 per 100,000 people, depending on the region. In Eastern Asia, for example, these ranges are much lower, with 1.05–18.6 per 100,000 and 4.6–57.3 per 100,000, respectively. Crohn’s disease is a chronic inflammatory disease that most commonly involves the terminal ileum and colon but can also involve regions located throughout the gastrointestinal tract [136]. Inflammatory responses are associated with T helper type 1 (Th1) and Th17 cells. In Crohn’s disease, the lesions do not involve the whole organ, and it is characterized by the occurrence of remissions and relapses. Another major inflammatory bowel disease is ulcerative colitis [137], a condition that affects only the colon. Similar to Crohn’s disease, patients also experience remissions, but the inflammatory responses are associated with the activity of T helper type 2 (Th2) cells.

CXCL1 has important functions in the course of inflammatory bowel disease. Serum levels of CXCL1 are elevated in patients with Crohn’s disease [139,140,141,142] and ulcerative colitis [140,141], where levels of CXCL1 are higher in ulcerative colitis than in Crohn’s disease [140]. Plasma CXCL1 levels are also elevated during relapses [140]. Elevated CXCL1 expression has also been found in mucosal biopsies in patients with Crohn’s disease [143] and ulcerative colitis [143,144,145,146], again with a higher expression of CXCL1 in ulcerative colitis than in Crohn’s disease [143,144]. CXCL1 expression in normal mucosa is expressed mainly in crypts [147]; however, in inflammatory bowel disease, CXCL1 expression occurs in pericrypt myofibroblasts [148], epithelial cells [147,148,149], enteroendocrine cells [150] and in inflammatory cells such as macrophages [148] and granulocytes [149].

CXCL1 may be a biomarker for ulcerative colitis [146]. Analyses of changes in gene expression indicate that CXCL1 is a hub gene in ulcerative colitis [151,152]. Additionally, mucosal biopsies from patients with ulcerative colitis show higher expression of another CXCR2 ligand: CXCL8/IL-8 [144,149], which indicates that CXCL1 does not act alone but rather with other CXCR2 ligands. Additionally, in a study on perfusates from patients with demonstrated elevated levels of CXCL1 [149], the concentration of CXCL1 was 3 times higher than CXCL8/IL-8, which indicates that in ulcerative colitis, CXCL1 has a major function among CXCR2 ligands.

CXCL1 expression depends on a variety of factors occurring in the inflamed intestines. For example, elevated expression of IL-17 [153] and TNF-α [154] in biopsies from colon mucosa of patients with inflammatory bowel disease showed elevated CXCL1 expression [155,156,157], dependent on the synergistic effects of those two cytokines [156,157]. Elevated CXCL1 expression in the gut of patients with inflammatory bowel disease may also depend on other factors such as interferon-stimulated gene product 15 (ISG15)/ubiquitin cross-reactive protein (UCRP) [158], IL-36α and IL-36γ [159], and the influence of gut microbiota [150,160]. In addition, studies on mice indicate that adenosine may also be responsible for increased CXCL1 expression [161].

CXCL1 causes chemotaxis and infiltration of inflammatory response sites by neutrophils in inflammatory bowel disease [139,148,162]. At the same time, CXCL1 is not the only neutrophil chemoattractant in inflammatory bowel disease. No less important may be CXCL8/IL-8 [148,162] and ISG15/UCRP [158,163]. The role of neutrophils in disease mechanisms in inflammatory bowel disease is unclear. Studies on dextran sulfate sodium (DSS)-induced colitis in mice have shown that these cells contribute to mucosal damage [164]. Neutrophils contribute to intestinal tissue damage in inflammatory bowel disease by producing ROS; CXCL1 also increases superoxide anion generation by these cells [148]. Neutrophils can also activate T cells [165], which are involved in inflammatory responses in inflammatory bowel disease. In the gut in paediatric patients with inflammatory bowel disease, neutrophils produce interleukin-23 (IL-23) [166] and are the main source of this chemokine in the gut in these patients. In contrast, in adult patients with inflammatory bowel disease, macrophages and dendritic cells are the main sources of IL-23 [166,167], which activates the IL-23→IL-17 pathway, important in the induction of inflammatory responses [168]. Increased levels of IL-17 contribute to the imbalance between Th17 cells and regulatory T cells (T_reg_) and the activation of pro-inflammatory Th17 cells. Via this axis, neutrophils may also influence inflammatory responses in adult patients with inflammatory bowel disease by producing IL-17 [165]. However, neutrophils can also help to alleviate the symptoms of inflammatory bowel disease and reduce inflammatory reactions in the gut, as demonstrated in animal experiments [169,170]. This is because neutrophils produce vascular endothelial growth factor (VEGF), interleukin-10 (IL-10) and anti-inflammatory eicosanoids [165].

### 3.10. Ileum and Contraction Amplitude

CXCL1 is also important in normal bowel function. Various diseases, such as endotoxemia [171], brain injury [172] and severe injury trauma [173], are associated with an increase in CXCL1 levels in either the blood or the gut. This leads to a decrease in contraction amplitude in the ileum [173], which results in feeding intolerance and ileus.

### 3.11. Obesity and Overweight

In America, an estimated 64% of the population is overweight, with 28% of the population being obese [86]. In Europe, the rates are only slightly better −60% and 23%, respectively [86]. Obesity is associated with chronic inflammation and high expression of CXCL1 in omental adipose tissue [174] and blood [175]. Additionally, animal studies have confirmed an increase in KC levels in adipose tissue [176] and blood [83] in obese mice compared to normal-weight mice. However, the role of CXCL1 in obesity-related diseases in humans is poorly understood.

One line of research is to analyze the association of CXCL1 with diabetes in obese individuals. CXCR2 ligands are produced in peri-pancreatic adipose tissue, as shown by experiments in obese rats [84], although young and old obese rats show no increased expression of the described chemokines. CXCR2 ligand expression in peri-pancreatic adipose tissue, as well as elevated blood levels of these chemokines, lead to the exposure of pancreatic islets to CXCR2 ligands. This leads to decreased islet function [83], which is followed by hyperglycemia. However, research in rats and on the INS-1 rat insulinoma cell line has not shown a direct effect of CXC chemokines on islet function [84].

## 4. Kidney

### Kidney Transplantation and Ischemic-Reperfusion Injury

Kidney transplantation is associated with renal hypoxia, caused by the disconnection of this organ from the donor’s bloodstream. Following transplantation, blood flow resumes, which oxidizes the transplanted organ, but similar to liver transplant, ischemic-reperfusion injury can occur, associated with increased CXCL1 expression in the kidney [177]. This effect is dependent on caspase activation, indicating that renal cell apoptosis may play an important role in the induction of CXCL1 expression [178]. The complement system is also an important factor in the induction of CXCL1 expression in transplanted kidneys, specifically the C3a receptor (C3aR) [179]. This effect is dependent on the activation of NF-κB by this receptor; the C5a receptor (C5aR) may also be responsible for this process [180], but a study on mouse proximal tubular epithelial cells did not show any effect on CXCR2 ligand expression [179]. Increased expression of CXCL1 is followed by infiltration of the transplanted kidney by neutrophils, as confirmed by studies in mice and rats [178,181,182,183,184,185]. These cells are responsible for kidney ischemic-reperfusion injury [185]; they release granules containing proteases, including cathepsin G, which contributes to damage to the transplanted kidney [183].

## 5. Reproductive System, Developmental Biology, Placenta, Pregnancy, Birth

### 5.1. Involvement of CXCL1 in Human Fetal Development and Placental Function

CXCL1 is important in human prenatal development. During the peri-implantation period, the trophoblast secretes IL-1β [186,187], which causes the activation of NF-κB in decidual cells. This is followed by the expression of CXCL1 in these cells [187], a chemokine that participates in implantation and placentation. CXCL1 is also involved in decidual angiogenesis [188] and causes granulocytic-myeloid-derived suppressor cell (G-MDSC) accumulation, which is necessary for the formation of pregnancy tolerance [189]. In the further stages of pregnancy, CXCL1 is produced by placenta cells [190,191], especially by human placental fibroblast-like cells [192,193]. It affects human pluripotent stem cells (hPSCs) [190,191,192,193] and causes long-term self-renewal of hPSCs; along with decreased cell-cell contact, this triggers neuronal differentiation [192]. This is why CXCL1 has been suggested as one of the components of the culture medium for hPSCs [193] and induced pluripotent stem cells (iPSCs) [190,191]. At the same time, decreased CXCR2 activation causes differentiation of hPSCs to the mesoderm and endoderm [190]. Nevertheless, due to bioethical aspects, the importance of CXCL1 in hPSC differentiation in human prenatal development has been poorly studied.

CXCL1 may also be important in the following stages of human prenatal development. CXCL1 levels are higher in cord blood plasma in preterm neonates than in healthy full-term neonates [194]. These CXCL1 levels are higher than in the plasma of adult individuals [195]. Additionally, blood levels of CXCL1 in preterm neonates are higher than in healthy adult individuals [195]. It is likely that CXCL1 is produced by the placenta [190,191,192], where it participates in angiogenesis [187] and pregnancy tolerance [189]. From there, it enters the fetal blood. However, the exact function of CXCL1 in human fetal development is still not known.

CXCL1 also undergoes increased expression in the myometrium during the period immediately preceding birth [196], in a process dependent on specific transcription factor avian v-maf musculoaponeurotic fibrosarcoma oncogene homolog F (MAFF) [197]. Although the exact function of CXCL1 during labor is not known, the dependence of the expression of this chemokine on a specific transcription factor indicates that it may be crucial in childbirth.

### 5.2. Endometriosis and Adenomyosis

Endometriosis is a disease in which endometrium is found at extrauterine sites [198]. Endometriosis affects approximately 11% of women of reproductive age. To date, due to many theories in circulation, there is an ongoing debate about the pathogenesis of endometriosis, but it is generally agreed that one of the elements of this disease is inflammation. Another gynecological disease is adenomyosis. It involves the presence of endometrial glands and stroma in the uterine myometrium [199]. It is estimated to affect 20%–25% of women, particularly women over 40 years of age. With that said, patients with endometriosis are more likely to also develop adenomyosis [200].

CXCL1 levels in the endometrium are elevated in patients with endometriosis [201,202] and adenomyosis [203]. On the other hand, blood CXCL1 levels in patients with endometriosis are not altered [204]. Additionally, an increase in CXCL1 expression in the endometrium appears to only occur in some patients [205]. In contrast, CXCL8/IL-8 expression is elevated in all cases [205,206]. Both chemokines act through the same receptor, and so it appears that CXCL8/IL-8 plays a major role in the aforementioned diseases, but CXCL1 may also be involved to some extent.

Elevated CXCL1 expression in the endometrium is due to pro-inflammatory cytokines [201] as well as interleukin-17A (IL-17A) [207] and VEGF [203,208]. CXCL1 causes infiltration of the endometrium by neutrophils, which participate in inflammatory responses. Additionally, CXCL1 can cause endometrial infiltration by MDSCs, which cause endometriotic growth and angiogenesis [209]. Paired with CXCL1, this can also directly cause angiogenesis in the endometrium [203]. Notably, the latter process occurs in adenomyosis.

## 6. CXCL1 as a Therapeutic Target

Following on from the first part of this paper [3], the most important direction for further research should be the application of the quoted data in practice. The CXCL1→CXCR2 axis plays an important role in the pathogenesis of many noncancerous diseases, especially in the discussed gastrointestinal diseases. The use of drugs directed against this axis may be an interesting addition to existing therapies. These drugs already exist on the market, e.g., CXCR2 inhibitors such as SB225002 [210], SX-682 [211], and NVP CXCR2 20 [212]. Some of the CXCR2 inhibitors are also CXC motif chemokine receptor 1 (CXCR1) inhibitors [211], which allows reducing the effect of CXC chemokines that are ligands for CXCR2 and CXCR1, including CXCL8/IL-8. Moreover, some postulate the use of anti-CXCL1 antibodies [213], which only cause the neutralization of CXCL1, although this approach is recommended mainly for tumorigenic processes, with a much greater body of research on the specific role of CXCL1 in cancer.

The use of the aforementioned compounds, together with the currently used therapies, may improve the therapeutic methods for the discussed gastrointestinal and abdominal diseases. On the other hand, this approach can be problematic. It is true that CXCL1 causes infiltration by neutrophils, which do cause tissue destruction, and a reduction in neutrophil infiltration can contribute to uncontrollable pathogen infection, which may worsen the patient’s condition. Another limitation associated with the use of CXCL1-reducing drugs is that most diseases are detected at an advanced stage, which means that the affected tissue has already been damaged or irreversibly destroyed, and so the use of CXCL1-reducing drugs will only inhibit the further progression of the disease.

## 7. Perspectives for Further Research

Although the significance of CXCL1 in gastrointestinal and abdominal diseases is well-established, little is known about the interplay between CXCL1 and other CXCR2 ligands (the human genome contains seven different CXCR2 ligands, including CXCL1), whose expression also increases. The vast majority of experimental work examines only one CXCR2 ligand, namely CXCL1 or CXCL8/IL-8, and so the role of the other six CXCR2 ligands remains unknown. There is also a question of why evolution in mammals (humans, mice or rats) has produced such a large number of CXCR2 ligands with identical properties. Some answers may lie in the mechanism of regulation of the expression of individual CXCR2 ligands, where each of their promoters differs slightly from others, which suggests some differences in function. Nevertheless, this is a matter for future experimental studies.

## Figures and Tables

**Figure 1 ijms-23-07151-f001:**
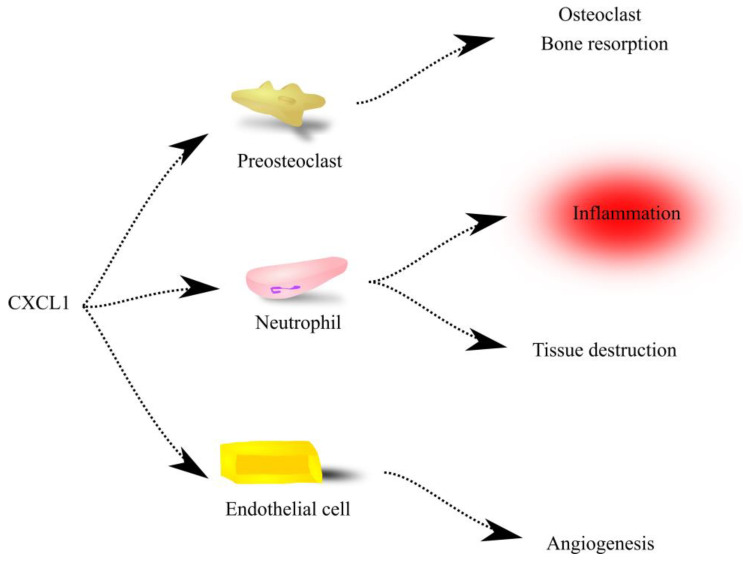
CXCL1 as a chemoattractant. Through its receptor CXCR2, CXCL1 causes infiltration by various cells into the sites of its high concentrations. In particular, this concerns neutrophils—cells that are involved in inflammatory reactions and may cause tissue destruction. CXCL1 is also a chemoattractant for preosteoclasts and causes osteoclastogenesis; high expression of CXCL1 in bone tissue induces an increase in the number of osteoclasts, which leads to bone resorption. Finally, CXCL1 acts on endothelial cells, which leads to their migration and results in angiogenesis.

**Figure 2 ijms-23-07151-f002:**
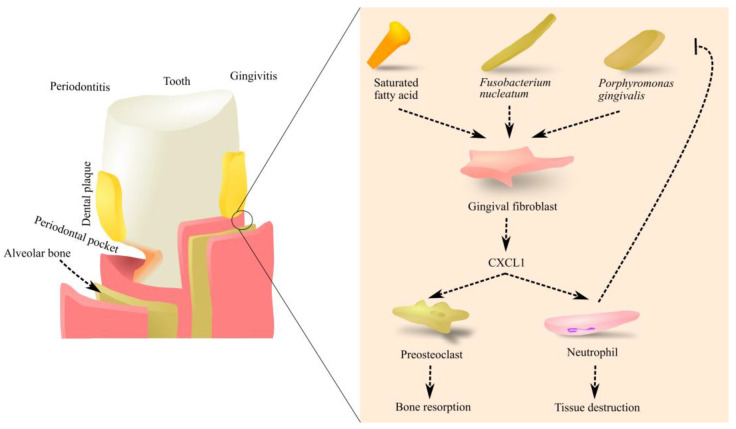
The role of CXCL1 in the formation of periodontitis. When teeth are not sufficiently cleaned, dental plaque is formed. It contains the bacteria *Fusobacterium nucleatum* and *Porphyromonas gingivalis*, which cause an increase in CXCL1 production by gingival fibroblasts. This effect is increased by a fatty diet and saturated fatty axis. CXCL1 causes the recruitment of neutrophils, which destroy microorganisms. On the other hand, neutrophils also contribute to tissue destruction. CXCL1 also acts on preosteoclasts, causing their migration and osteoclastogenesis. This leads to alveolar bone resorption. As a consequence of these processes, periodontal pockets and periodontitis are formed. Abbreviations: CXCL1—CXC motif chemokine ligand 1. Source: own elaboration.

**Figure 3 ijms-23-07151-f003:**
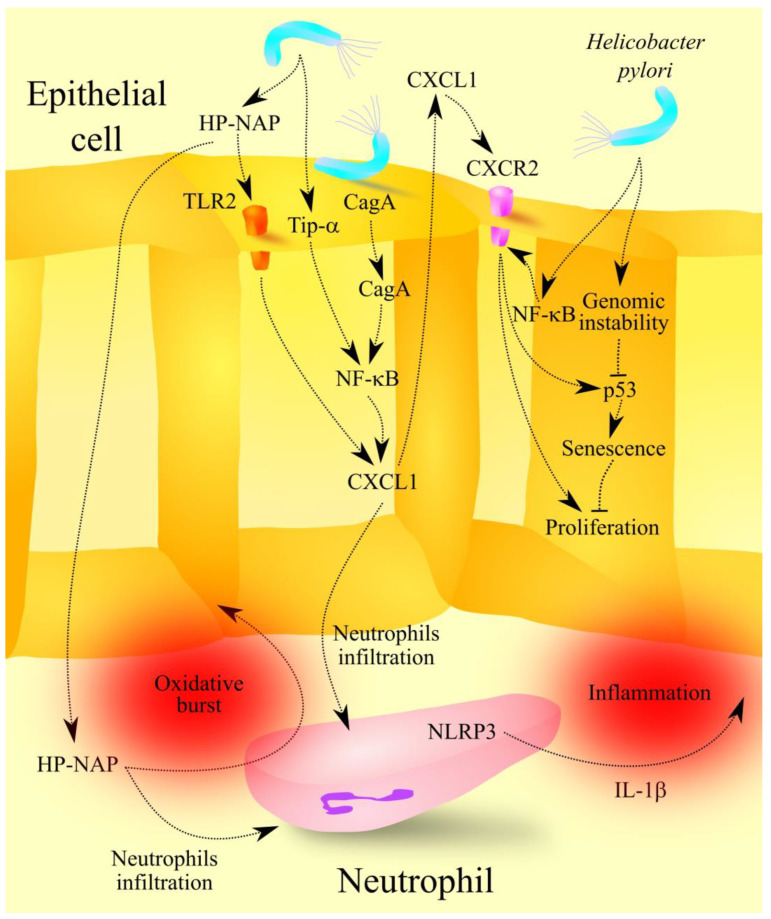
The role of CXCL1 in the molecular mechanisms of *H. pylori* infection. *H. pylori* increases CXCL1 expression in the epithelial cells of gastric mucosa due to virulence factors Tip-α, HP-NAP and CagA, which activate NF-κB and TLR2. Then, CXCL1 and HP-NAP cause infiltration of the gastric mucosa by neutrophils. These cells cause chronic inflammation through the secretion of IL-1β and ROS. CXCL1 also causes senescence of gastric mucosa by activating the CXCR2 receptor and increasing p53 activity. This process leads to gastric mucosal atrophy. At the same time, by activating NF-κB, *H. pylori* increases CXCR2 expression and thus sensitizes cells to CXCL1 activity. *H. pylori–*induced senescence is a state in which cell proliferation is inhibited. On the other hand, *H. pylori* also causes genomic instability, which leads to mutations in the *TP53* gene. This reduces the activity of the product of this gene: p53. Because CXCR2 activation also causes inactivation of p53, it leads to the proliferation of gastric mucosa cells and the formation of gastric cancer associated with *H. pylori* infection. Abbreviations: CagA—cytotoxin-associated gene A; CXCL1CXC motif chemokine ligand 1; CXCR2—CXC motif chemokine receptor 2; HP-NAP—*H. pylori*-neutrophil activating protein; IL-1β—interleukin-1β; NF-κB—nuclear factor κB; NLRP3—nucleotide-binding domain and leucine-rich repeat related family, pyrin domain containing 3; Tip-α—TNF-α inducing protein; TLR2—toll-like receptor 2. Source: own elaboration.

**Figure 4 ijms-23-07151-f004:**
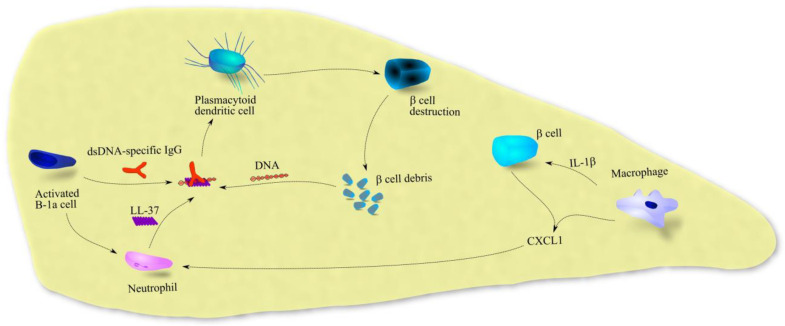
The importance of CXCL1 in T1DM. In patients with T1DM, during inflammatory reactions in the pancreas, CXCL1 expression is increased in macrophages and influenced by IL-1β in β cells. This chemokine causes the recruitment of neutrophils to the pancreas. In this organ, neutrophils under the influence of activated B-1a cells secrete LL-37. This peptide with dsDNA and dsDNA-specific IgG forms a complex that activates plasmacytoid dendritic cells (pDC), which initiates the destruction of pancreatic islets. Abbreviations: CXCL1—CXC motif chemokine ligand 1; IL-1β—interleukin-1β. Source: own elaboration.

**Figure 5 ijms-23-07151-f005:**
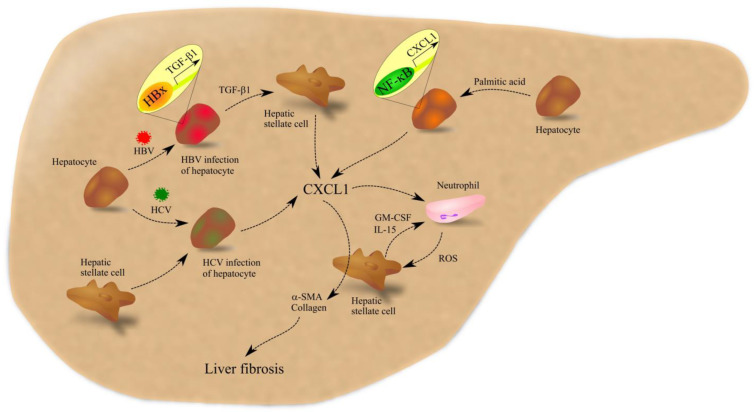
Involvement of CXCL1 in liver fibrosis. In the liver, CXCL1 is secreted by HCV-infected hepatocytes. This effect is enhanced by HSC. CXCL1 is also secreted by hepatocytes that contain too many saturated fatty acids as a result of an abnormal diet. In addition, HBV infection of hepatocytes increases the expression of TGF-β1 in these cells, a result of the direct action of HBx viral protein on the promoter of the *TGFB1* gene. TGF-β1 increases the expression of CXCL1 in HSC. The aforementioned processes increase CXCL1 levels in the liver. This chemokine causes the recruitment of neutrophils to the liver. These cells secrete ROS, which causes activation of HSC, which secretes IL-15 and GM-CSF, which increase the viability of recruited neutrophils. Under the influence of CXCL1, HSC increases the expression of α-SMA and collagen, which leads to liver fibrosis. Abbreviations: α-SMA—α-smooth muscle actin; CXCL1—CXC motif chemokine ligand 1; GM-CSF—granulocyte-macrophage colony-stimulating factor; HBV—hepatitis B virus; HBx—hepatitis B virus X antigen; HCV—hepatitis C virus; IL-15—interleukin-15; NF-κB—nuclear factor κB; ROS—reactive oxygen species; TGF-β1—transforming growth factor β1. Source: own elaboration.

## Data Availability

Not applicable.

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
