# Peer review of "The Importance of CXCL1 in the Physiological State and in Noncancer Diseases of the Oral Cavity and Abdominal Organs"

_ijms, 2022, doi:10.3390/ijms23137151_

Round 1
Reviewer 1 Report
The manuscript by Korbecki et al. reviews the functions of the chemokine CXCL1 in disease. The review is comprehensive and well put together. Overall, it is worth publishing. I only have two suggestions:
1. It would help the broad audience of IJMS if the authors include an image illustrating the general function of CXCL1 as a chemottractant along with a brief discussion.
2. Because of the role of the CXCL1/CXCR2 axis in disease, the authors correctly point out that its therapeutic targeting is important. However, this is only a very brief statement in the Conclusions. I suggest expanding this into a separate section describing the up-to-date knowledge on CXCL1 therapeutics. I the new Conclusions section, the authors could focus on the new future directions of the research to address key questions about CXCL1 function in disease and development of novel CXCL1 inhibitiors.
Author Response
Rev.1
We are very grateful for the very kind and insightful review of our manuscript.
The manuscript by Korbecki et al. reviews the functions of the chemokine CXCL1 in disease. The review is comprehensive and well put together. Overall, it is worth publishing. I only have two suggestions:
- It would help the broad audience of IJMS if the authors include an image illustrating the general function of CXCL1 as a chemottractant along with a brief discussion.
The figure showing the general properties of CXCL1 along with the chemotaxis properties of the cells described in this paper has been added
- Because of the role of the CXCL1/CXCR2 axis in disease, the authors correctly point out that its therapeutic targeting is important. However, this is only a very brief statement in the Conclusions. I suggest expanding this into a separate section describing the up-to-date knowledge on CXCL1 therapeutics. I the new Conclusions section, the authors could focus on the new future directions of the research to address key questions about CXCL1 function in disease and development of novel CXCL1 inhibitiors.
The final section has been expanded as recommended by the reviewer. New conclusions have been written suggesting the future direction of CXCL1 research

Reviewer 2 Report
The review manuscript entitled “The importance of CXCL1 in the physiological state and in noncancer diseases of the oral cavity and abdominal organs,” attempted to summarise the research knowledge regarding the CXCL1 chemokine. Overall, the manuscript is well written and contains a plenty of informations. The followings are the points which I would like to suggest the authors to correct.
1. The manuscript comprehensively describes the importance of CXCL1 in the body. The title of the manuscript might be changed as follows.
“The importance of CXCL1 in the physiological state and in noncancer diseases of the body”.
2. In every figure, the structure of each subjects are quite unclear.
Especially in Fig. 2, Heliconacter pylori is obscure because it was drawn in light green in yellow background. The shape of the epithelial cell is not appropriate. The extracellular and intracellular border (cell membrane) should be clearly drawn.
In Fig. 4, letters for NF-B and HBx are too small.
3. What is 3c diabetes?(230) Should be explained.
4. It looks like that “With that said (273)” is a spoken language.
5. “For this reason, there is a mechanism that inhibits that excessive response.(410)
“That” might be “the”.
Author Response
We are very grateful for the very kind and insightful review of our manuscript.
Rev.2
The review manuscript entitled “The importance of CXCL1 in the physiological state and in noncancer diseases of the oral cavity and abdominal organs,” attempted to summarise the research knowledge regarding the CXCL1 chemokine. Overall, the manuscript is well written and contains a plenty of informations. The followings are the points which I would like to suggest the authors to correct.
- The manuscript comprehensively describes the importance of CXCL1 in the body. The title of the manuscript might be changed as follows.
“The importance of CXCL1 in the physiological state and in noncancer diseases of the body”.
In our previous work, we described the importance of CXCL1 in nervous tissue, bone and muscle. In this manuscript, we have described the importance of CXCL1 in the gastrointestinal tract and abdominal organs (kidney, female reproductive system, pregnancy). In this study, we did not write about the importance of CXCL1 in nervous tissue, therefore the topic of the article pointing to the abdominal cavity indicates the content of the article in more detail.
- In every figure, the structure of each subjects are quite unclear.
Especially in Fig. 2, Heliconacter pylori is obscure because it was drawn in light green in yellow background. The shape of the epithelial cell is not appropriate. The extracellular and intracellular border (cell membrane) should be clearly drawn.
In Fig. 4, letters for NF-B and HBx are too small.
Figures 2 and 4 have been corrected as recommended by the reviewer
- What is 3c diabetes?(230) Should be explained.
The description of 3c diabetes has been developed
- It looks like that “With that said (273)” is a spoken language.
- “For this reason, there is a mechanism that inhibits that excessive response.(410)
“That” might be “the”.
Linguistic errors have been corrected
